# Evolutionary System Design for Virtual Field Trip Platform

Hyodong Ha [1] and Changbae Mun [2,*]

1 Department of Big Data, Hanyang Women's University, Seoul 04763, Republic of Korea; special007@hywoman.ac.kr
2 Department of Electrical, Electronic & Communication Engineering, Hanyang Cyber University, Seoul 04764, Republic of Korea
* Correspondence: changbae@hycu.ac.kr

**Abstract:** This study proposes a systematic design methodology for supporting field trips in software education. Various attempts have been made to enhance the educational effects of field trips, but it is difficult to continue them due to the limitations of educational resources. Virtual field trips (VFT) have emerged to supplement this, but they are still in need of development. This study designed a platform that integrates the VFT functions of each school's educational system into one, which guarantees each system's unique activities and supports the course of technological progression. This proposes a structure in which these processes continue to evolve, and the platform learns from this data to facilitate the evolution of the technological ecosystem. The system design methodology derived from this research can be applied not only to the educational domain but also to various industrial fields.

**Keywords:** system design; SW education; virtual field trip





## 1. Introduction

The Fourth Industrial Revolution, first mentioned by Klaus Schwab, is currently underway [1]. That humanity is currently living in an era of technology and data is not an exaggeration. The convergence of technologies like big data and artificial intelligence is influencing employment and changing the industrial ecosystem [2]. In line with these changes, information systems are continuously evolving in different ways. For example, in the early stages of the development of the Internet of Things, connection and control were central, but software algorithms are gradually moving toward optimization and autonomous operation in terms of performance improvement and prediction [3]. In addition, systems must provide functionality and complexity via growth and change as domains change at the software level; therefore, architecture is also being applied and developed with evolutionary mechanisms [4], which refer to a methodology aimed at maintaining progressive design within a system. It involves a process of iterative improvements according to formalized analysis procedures, using diagrammatic tools [5].

Despite the development of these technologies, the educational information system has failed to adapt itself to environmental changes and demonstrated limitations. Numerous schools were temporarily closed during the COVID-19 pandemic, and existing offline classes were suddenly converted to online classes without sufficient action strategies in place [6]. Most educational activities in various Asian countries, including Hong Kong, Indonesia, and South Korea, have transitioned to online formats [7–9]. Offline field learning has either been canceled or switched to an instructor-centric online method [10]. The abrupt change also led to questions about the effectiveness of the class. In the case of online/recorded classes, absence was not a problem for students with high learning motivation, but it was difficult for students with low learning motivation to continue the class [11]. And students with computer anxiety faced difficulties in using the system; therefore, the system could not meet the students' learning requirements [12].

Meanwhile, there have been ongoing studies attempting to apply virtual reality in educational settings. Makransky and Mayer (2022) confirmed that immersive technologies using head-mounted displays make education feel lifelike, drawing students into enjoyment and interest, and leading to a positive learning experience [13]. When immersive virtual field trips (VFT) are conducted based on appropriate educational design, they can induce intended behavioral changes in students [14]. Additionally, research targeting elementary school students showed that immersive VFTs can help reduce test anxiety [15]. Furthermore, in the study by Meed et al. (2019), adaptive feedback was integrated into VFTs to automatically provide the necessary information for individualized learning [16]. Moreover, educational applications using augmented reality have yielded excellent results in terms of the quantity, complexity, and reliability of activities, compared to the control group, when tailored to students' levels of knowledge [17].

Ultimately, just as an information system is defined as a system that interacts to provide the necessary information to support an organization [18], an educational information system should be able to support integrated education with sustainability as its goal from the perspective of the virtual reality ecosystem. The virtual reality ecosystem consists of content, platforms, networks, and devices [19]. Previous studies [13–17] have focused on educational content and devices. In contrast, our study focuses on a platform that connects various stakeholders online and allows for the supply of educational content by integrating existing legacy systems. The platform evolves, and its features continually reconfigure through systematic feedback. For example, even in a basic web board with just four fundamental functions—input, edit, delete, and update—refined features are updated through various user characteristics. These updates accumulate and lead to the development of a unique system for the platform. Additionally, it is not just one element in the system that evolves independently; the development of each element is considered alongside system requirements in terms of content, network, and devices to have a positive impact on the overall platform. Therefore, from the perspective of overall components within the platform, we can keep pace with the speed of technological advancements. In this study, we explore methodologies and evolutionary modeling, which refer to a methodology aimed at maintaining progressive design within a system. It involves a process of iterative improvements according to formalized analysis procedures, using diagrammatic tools [20] design techniques for systematically providing experiences similar to offline experiences in an online field learning platform, and we confirm the feasibility of online platforms for education and experiential programs centered around various stakeholders. Software architecture is not a fixed entity; it includes methods that are continuously refined to achieve the desired outcome.

In software engineering, the advantages of an evolutionary process lie in its capacity for iterative improvements and speed. The greatest risk factor in software development is the need for large-scale modifications during the final stages. Evolutionary modeling has the attributes of 'immediate reflection of iterative improvements' and 'capability to utilize prototypes for decision making'. Over time, this approach enables the development of software that meets user satisfaction by continually improving its features. One drawback of this modeling approach is that it involves iterative development phases, making it challenging to estimate the overall resources that will be invested in the development. However, its utility is high in continuously improving the quality of the end product. This modeling approach proposes a comprehensive technological ecosystem that integrates various VFT systems and educational management information systems (EMIS), allowing these systems to function autonomously and accumulate performance data. Each system evolves its features based on this accumulated data. Evolutionary modeling is incorporated into software architecture. This architecture adapts to continually changing environments while maintaining functional stability. Microservice architecture has also been introduced from this perspective and has evolved into a methodology with high scalability when combined with the structural characteristics of system engineering in the technological ecosystem [4]. Additionally, in the case of large international airport systems, there is a

process that reflects historical data and execution outcomes in the system management and medium- to long-term planning [21]. Through this process, a virtuous cycle is systematically established, leading to ongoing improvements.

## 2. Information System: Need to Adopt Intelligent Systems in Education

Online education systems have been actively implemented as a mainstream approach in the current global education system environment. The Fourth Industrial Revolution has greatly influenced the convergence of educational information systems (EISs) with digital technologies. In the past, the educational system had limitations in regard to providing the class management functions represented by learning management systems (LMSs). In the era of the Fourth Industrial Revolution, state-of-the-art technologies such as artificial intelligence, big data, and augmented reality–virtual reality have been introduced into EISs, leading to the development of educational platforms, devices, content, and services. Various technologies drive the digital transformation of higher education institutions, depending on social, organizational, and technological perspectives [22]. These advanced EISs focus on minimizing the differences between online and offline education and effectively providing personalized education online.

Owing to the outbreak of COVID-19, the education world went through three phases in 2020 and 2021: disruption, transition, and reimagining [23]. The disruption phase refers to a period of disruption in the education world, when offline education became unavailable and online education emerged into the mainstream as an alternative. The outbreak of COVID-19 caught everyone off guard. However, much like how software development outcomes differ depending on the level of capability maturity model integration (CMMI) [24], organizations showed significant differences in their responses based on the maturity level of their digital technology. During this period, policy and technical discussions regarding online education took place. These discussions centered around various challenges such as online class format, class management, content utilization, and curriculum maintenance. The transition phase corresponded to the full-scale adoption of the online curriculum. In this phase, the effectiveness of the education and the quality of the content were discussed. For instance, despite being willing to participate in online classes, students still showed a preference for face-to-face classes or felt that they were inferior [25]. The reimagining phase refers stage after the second half of 2021, when researchers considered the growth of online education in new environments. In this phase, educational institutions began to identify individual student-specific online education methods. Since participation is a key determinant in learning, we made various attempts to offer students virtual real-world experiences, museum and gallery tours, simulations, and sandbox environments [23]. While undergoing these three phases, the education world examined how to improve the quality of the content in EISs and discussed new and innovative education techniques.

Along the way, various LMS companies have gradually developed more technologically advanced online course features.

The global LMS market size was estimated at USD 17.36 billion in 2022 and is expected to reach USD 69.69 billion by 2030, with a compound annual growth rate of 19.2% between 2023 and 2030 [26].

The leading global LMSs include Canvas and Moodle. LMS is categorized into open source and commercial; in the case of open source, it can be modified according to the educational institution's needs to reflect the organization's unique characteristics [27]. In South Korea, many institutions have built their own respective systems based on the specific educational situations of their schools. The advantages of building their own systems are recognized in various aspects. The LMS market accounts for a significant portion of the Internet service industry market and is also growing to a significant size in the software-as-a-service (SaaS) market [28].

The scope of the functionality provided by LMSs has gradually evolved. Today's LMSs are used to automate the management of curriculum, education, learning, and development programs within an educational institution; they are also used to provide

statistical information. They are usually offered as a service on a paid subscription basis. In the post-COVID-19 era, the education market has increasingly used SaaS in the context of cloud processing. Conventionally, in the EIS field, it was generally difficult to develop functions, invest money in servers, and provide innovations in services.

Presently, in contrast, innovations are currently being developed across the entire education system based on the cloud. The LMS functions to technically manage the interactions between professors, students, and administrators in the physical space of a school. Furthermore, an ecosystem is built across educational activities such as the interactions between professors, students, and administrators, as well as between parents and local organizations. In an EIS, it is important for the LMS to support these three types of activities to proceed in a stable fashion. In this context, a variety of tools have been developed to improve the in-class performances of students.

However, classes that simply deliver online lecture videos have many limitations. One of them is the lack of student participation in the activities performed in the class. A prime example of this is when a student plays a live or video lecture but takes no action [29]. Le (2022) comparatively analyzed recorded and real-time methods like Zoom in terms of academic performance. The findings suggest that students in the lower 50% performed relatively worse in recorded classes [30]. In offline classes, interactions may occur with nearby students, but in online learning, the class ends without any interaction. Various technological attempts have been made to overcome this limitation of EISs. Since the mid-2000s, interactive features have been installed in EMISs and LMSs as countermeasures. For example, devices proceed with the lecture only when the student provides feedback. Nevertheless, students have continuously skipped this process by simply clicking on the lecture content without paying attention to the EIS lecture. In the end, to provide meaningful results, technological attempts must be made to increase student participation in online education and time must be provided for direct experiences with the lecture in the system to appropriately apply this technology.

This study discusses a methodology from the perspective of a field trip in the field of information system software, to which the authors belong. In Section 3, we propose a more precise methodology design technique for systematically supporting students' experiential learning. And we compare this proposed methodology with conventional software design methodologies. The characteristics of the design methodology proposed herein are analyzed in comparison with the results from previous studies.

In this study, we propose a VFT design methodology following the flow outlined below, aiming to explore the evolutionary perspective in the field of information system software.

(1) In the Introduction, we first discuss system development methodology. We present the development process and functional changes of online education systems. We explore approaches for delivering learning experiences in the system to achieve educational goals.

(2) In Section 3, we propose more precise methodology design techniques to systematically support experiential learning for students. The platform operates as a single technological ecosystem, and the system undergoes continuous development to sustain educational activities. We analyze the interactions of data emerging from these processes and incorporate the findings into the system. Within the framework of platform policies, this cycle repeats, and the system evolves through interactions with other systems.

(3) In Section 3.3, we compare the VFT methodology with existing software design methodologies. Through usability tests conducted with five users, we analyzed the evolutionary elements and characteristics each system possesses within the technological ecosystem. From this perspective, we analyze the characteristics of the VFT methodology.

(4) In Section 4, we conducted an empirical analysis. As a result, we confirmed that user behavioral elements were additionally derived by more than 20% in the features designed using the VFT methodology.

## 3. Research Method

### 3.1. Field Trip and Education Information System

The software field has a vast array of technology domains; accordingly, students are limited in their abilities to directly experience areas they are interested in through their undergraduate studies. It is possible for students to build their knowledge through participation in conferences and extracurricular activities in their major field(s), but only in some areas and for limited periods of time. Similarly, regular periodic experiential learning for elementary and middle school students has various limitations in terms of implementing the periods and education. External education and experiential learning beyond the regular curriculum are difficult to properly manage offline. As a solution to these problems, various technological support approaches are being sought for online classrooms.

In this regard, EISs and LMSs offer considerably greater opportunities to improve the performance of experiential learning through their functionally advanced technologies. Therefore, it is necessary to limit the scope of the experiential learning that should be systematically functionalized and applied in the EIS. Experiential learning (including field trips) allows students to connect with a topic or concept through a hands-on experience. This type of learning includes practical learning outside classrooms, such as participation in experiments and research. In a self-directed experiential education model, students are in charge of their own learning. This helps them develop their own curiosity, collaboration, and self-motivation. In other words, experiential learning comprises a process in which students experience knowledge through external educational experiences and learn by building information based on their experiences. According to the Ministry of Education, creative experiential activities emphasize students' autonomous educational activities, allowing students and professors to discuss plans for educational activities and to divide roles to implement them.

Currently, however, EISs are limited in technically managing and supporting these activities. Many functions of EISs are limited to saving and listing simple histories. Furthermore, the off-campus experiential learning conducted by individual students in middle schools can be somewhat limited. Students participate in their own external activities after submitting applications to the school, but the process of application submission, report submission, and approval is very complex, in conjunction with the school administration. Therefore, functions are being adopted to resolve the inconveniences of students and parents and to simplify the administrative process from application to approval [31]. This study attempts to address the above limitations of EIS using an evolutionary modeling technique. In other words, it searches for a method to significantly improve an existing EIS to enhance students' educational experiences. In this process, we identify key functions that can be introduced into the system and verify their effectiveness.

In terms of data, the behavioral and experience information data of students are combined when considering the adoption of a module for adapting and learning on their own. In the South Korean education system, EISs have shown rapid social progress, especially since COVID-19. E-learning appeared in the early 2000s in the form of a video lecture platform, but the advancement of the EdTech industry utilizing big data, virtual reality–artificial intelligence, etc., remained stagnant. In early 2020, online classes were rapidly implemented across public education owing to the COVID-19 pandemic, and in the process, a social consensus was formed on the need to vitalize the EdTech industry. At this point, the forms of the corresponding educational platforms were transformed, evolving from one-way online education to online classes able to allow discussion while watching content, as well as to provide learning reports and mutual evaluation. In this ongoing process, it is necessary to actively foster the EdTech industry, link it with the existing education system, monitor new changes in terms of educational information technologies, and make this flow an opportunity for evolution and advancement.

### 3.2. Education System and Technology Ecosystem

It is necessary to build an ecosystem of systems based on this empirical statistical information. The primary users of EMISs are professors, students, and academic administration staff. After the semester begins, the EIS continuously accumulates the usage history data of these users. This study applies requirements engineering techniques based on the statistical experience information from these users. Examples of statistical experiential information are as follows. Strousopoulos et al. (2023) developed a mobile application system for displaying sculptures and performed content recommendations using fuzzy weighting [32].

From the students' perspective, experiential activities represent small subjects. From this perspective, each activity constitutes a virtual classroom where classes are conducted. Correspondingly, a system that encourages active participation in education and interactive responses to results and processes is required. On the teachers' side, it is necessary to provide personalized services for each subject, discuss each activity, and manage the history and activity information. From the perspective of school administrators, the system focuses on creating activities and managing their operations. That is, they can help in approving the creation of activities, managing statistical information, and managing history. These programs should be operated in cooperation with experiential education institutions to ensure that the experiential activities result in substantial experiential learning.

Overall, the system structure consists of education systems (one system for each institution), a data platform (for students), and an overall management system. The management system provides activity authentication, location verification, and activity registration functions for experiential activities (including field trips). The central management platform provides integrated information on the experiential activities from each institution's system and a process for managing activity information. In this process, information related to student participation, certification, and evaluation can be systematically managed in a common manner. From the institutional perspective, it enables the sharing of educational information and student activity details between institutions. Based on this aspect, the system will be able to contribute to improving student work efficiency and supporting the achievement of educational goals in terms of experiential learning.

Accordingly, the following proposals will be reflected in the system design.

Goal: To introduce an evolutionary methodology to the user learning information-based ecosystem. An evolutionary element refers to an individual feature that engages in progressive interactions within the system. The evolutionary method is the functionalization of this element within an independent object.

I.   Conventional learning management systems (EMISs and LMSs) focus on the systematization of student learning management and academic administration information.
II.  A data platform for students' educational activities is used to classify their histories and model application methods. In this process, students create learning profile data and educational activity histories.
III. The administrator of a group manages the usage for each school or educational institution to support quantitative educational goals for experiential activities such as content verification, educational certification, and performance management.
IV.  The system acts as an intelligent manager to achieve the common learning goals of each school's system. Each institution operates as an object of knowledge and information within the technology ecosystem.
V.   As a necessary element for the system's self-advancement, it supports decision making regarding educational activities and applies it to the education platform linkage system, thereby facilitating its functioning as an integrated system.

The techniques of various educational platforms in the LMS technological ecosystem are continuously discussed and the advantages of each system are statistically derived.

*3.3. System Design of the Education Platform for Virtual Field Trip (VFT)*

In general, school administrators and instructors manage the actual operations of academic affairs and classes through the EMIS. Student management and education administration tasks can be processed through an integrated education information portal system. Currently, in offline practical learning classes, portfolios can be registered and certified in connection with an external activity management system. However, the services provided by these systems are limited to simply saving and listing information.

Students' experiential activities must incorporate functions focused on the systematic and technical management of external activities. In particular, when designing experiential activities by dividing the functions required for web-based roles between professors and students, functions tend to be built by considering the respective behavioral characteristics of professors and students when dividing the roles of the users in the system. A personalized adaptive learning system is needed to provide customized educational information to students based on the basic information obtained by analyzing the learning patterns of each student. This can be accomplished by building functions in consideration of the role and work characteristics of the professor as an upper-level management entity for each student. A previous study argued that when adopting an information system to achieve the goals of a curriculum, extracurricular activities fall into the same category as creative experience activities and should be managed with the utmost importance in the school's information system. This is because extracurricular programs are also developed and managed based on students' academic data.

Each school's online courses have technically stable functions for providing educational information, checking attendance, providing assignments, and grading students online. Nevertheless, situations can occur where students have difficulty focusing on studying owing to technical or personal reasons, as shown in the following examples.

I.    Having difficulties using the system for VFT activities outside the classroom, depending on the system and network environment;
II.   Having difficulties in real-time level 2 or higher communication between the instructor and students during learning external activities of VFT;
III.  Having difficulties in interactions in forms other than video lectures (e.g., in cases of students requiring practical learning.

This study proposes a platform that can deploy various VFT EMIS separately within a technology ecosystem as a solution. The platform consists of four main process flows and two platform policies, with each flow accumulated on the platform, comparing EMIS action data with educational performance and ensuring an organic operation of results data.

Here are the four process flows in the VFT class support system:

I.    VFT system policy: This is to classify educational information;
II.   Define the flows of the actions by users (students, professors, and staff users);
III.  The platform's "self-learning" using big data in the education system;
IV.   Compare the same subjects within each system in the platform.

These process flows illustrate the flows of the functions performed by the platform's central server. In the cloud platform, each major system is identified as a node, and direct data processing is performed using access rights. Communication between these nodes is not allowed, and all communication is done step-by-step through the central application programming interface (API). The processing and learning involved in this process are defined in the platform policies.

Here are the platform policies from the point of view of supporting the VFT class support system:

I.    There are no restrictions on the spread of action data generated by each system;
II.   Nodes unique to each system have their own workflow, and they are connected to a central API;
III.  The flows of the nodes are maintained independently and standardized.

The primary nodes of each system are integrated to form an independent workflow that is interconnected with the EMIS application functionality to achieve procedural automation. In this connection, data are shared through statistically presented results. This approach ensures that data from each node is synchronized and learned step by step, resulting in continuous improvement. Decisions for key actions in the curriculum and systemic support are accumulated in the system as quantitative data.

Figure 1 shows the structure and flow of the entire system. The starting point of the educational administration process is the accumulation of information produced by students and professors. This paper focuses on data from the VFT perspective. From this point of view, information on each educational institution is created. In this way, the VFT information of each educational institution is standardized and processed within the platform. In the platform interface scheme, collection procedures are adopted to regularly transform the flow of information generated by each educational institution.

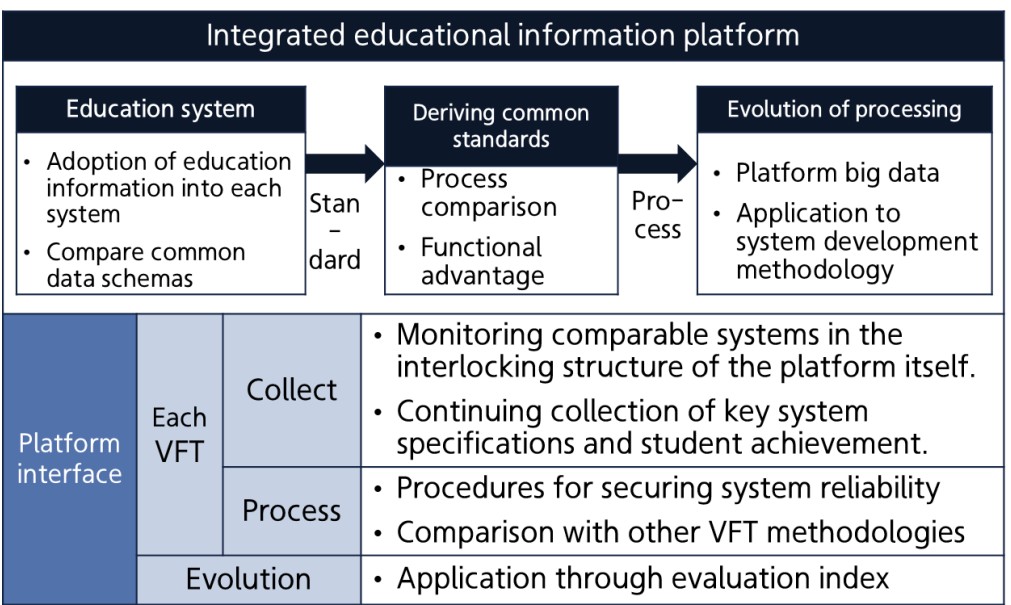

**Figure 1.** Overall process overview of the platform.

The third step is information processing for platform evolution. As a preliminary work, the big data collected from each VFT is classified according to the relationship between unit systems. In this process, each educational institution's system discriminates and recognizes each other's systemic advantages. Over time, the strengths of each system on the overall platform become functional. Also, these functional advantages are gradually being ported to each system as a whole. In this aspect, the platform's function list is reorganized based on the functions that have gained support from users. This process corresponds to the evolution of the system development methodology proposed in this paper. Therefore, this study proposes an evolutionary method that complements the traditional system design methodology [33,34]. Structural system analysis or object-oriented system analysis techniques have also been applied to various systems production for a long time. The authors studied techniques that can be applied to continuously evolving systems in specific technology domains. The authors compared the design methodologies used by many IT service companies and analyzed the company's system design best practices with two professional engineers. As a result, this study came to propose a system analysis methodology with academic value in the field of educational information systems, and systemic benefits are recognized in many aspects.

Figure 2 shows the data flow configured on this platform. From the point of view of state transition, it is a data flow diagram of the process in which students take classes such as VFT and Math. This figure includes state transition configuration in DFD level 0. Functionally, we prepared for exceptional data loss by including an interaction check

program. In addition, data flows through each process in each system and is converted according to the API. When a VFT execution request occurs as the first step in EMIS, activities for VFT are accumulated in each system within the platform. During this period, students take various classes such as math and history, and these data are also stored in a separate data schema. In this process, various experiential elements will be reflected in the VFT, and these functional advantages will be statistically analyzed in the system and reflected in the platform.

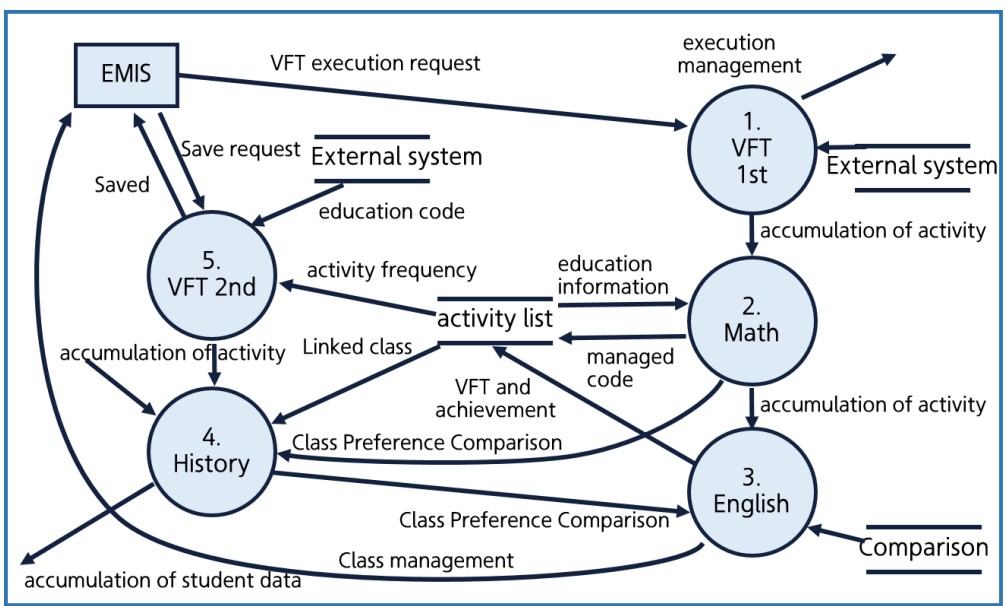

**Figure 2.** Data flow diagram for VFT class.

Figure 3 shows the relationship between ecosystems and systems. When functional design is applied to object-oriented programming (OOP), a factory method pattern is introduced [35]. In this process, we made it possible to create an instance directly from a functional subclass. Also, as an additional design pattern, an iterator pattern is assigned to each system node. So, it was designed to be applied when accessing each information in the form of defining an individual system as a structure [36,37]. The platform entered the technology ecosystem, and within it, communication between each LMS architecture was constructed. This constitutive element was prepared so that it could be expressed as an evolutionary element in the system.

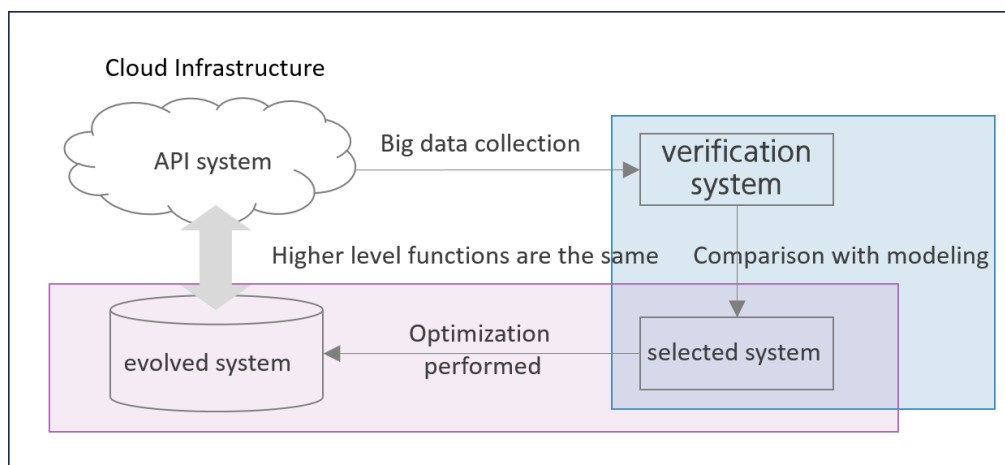

**Figure 3.** The relationship between the technology ecosystem and the platform.

A usability test for the platform of this study was conducted as a prototype. The research subject was limited to one person with a high understanding of the educational platform.

Subject: 2 middle school teachers, 2 university professors, 1 professional engineer.

The experimental method is comparison and analysis for functional review. In 2021, company A organized the functions based on the data used for system analysis and design. Teachers and professors confirmed the usability aspect of the VFT system and asked to present requirements. In the second stage, professors were asked to present their opinions by comparing the functional requirements with the existing projects. Professional engineers analyzed the data in terms of function design and requirements engineering. As a result, it was confirmed that more than 12% of design functions containing structural expressions were collected in addition to formal functional designs.

To perform a technical analysis of evolutionary modeling, we introduced content analysis to the existing system requirement analysis results. This method is a qualitative research approach from the perspective of instructors; its greatest advantage is that it allows for statistical analysis of the text data collected. The process was conducted in two stages. In the first stage, we analyzed the requirements collected for general EMIS and LMS. The second stage involved gathering requirements from feedback generated from an evolutionary modeling perspective after its application. We added explanations about the nature of the questions and the content analysis. Through this description, the justification for the methodology has been further strengthened.

The four criteria for collecting requirement analysis data are as follows.

Criterion 1: Is this feature similar to the educational effectiveness of a field trip? Y/N (Y: Q.2a, N: Q.2b).

Criterion 2.a: What is the most similar feature between FT and VFT in terms of effectiveness?

Criterion 2.b: What is the feature that most needs improvement in VFT?

Criterion 3: What additional features are needed for application across various learning processes?

Criterion 4: What features have been identified through the evolutionary functionality of VFT?

We refined the analysis procedures in accordance with the definitions of content analysis for the requirement data. Based on the techniques of content analysis for requirement analysis, we refined the information corresponding to qualitative data. Next, during the sense-making process, we divided the data into units and categorized them according to core consistencies and meanings, generating descriptive evaluation indicators. Through this process, we are able to analyze the inherent characteristics of this methodology.

## 4. Conclusions and Discussion

Empirical analysis has led to the discovery of further user behavioral factors within this design methodology.

We divided the requirement analysis text into sentences, performed pre-processing, and carried out a structured content analysis of the data. We analyzed all the clauses in the analytical data and identified 286 characteristics, 254 basic functions, and 63 areas for improvement. After conducting a pre-test on the result data, we removed any outliers. We categorized the data by core modules and conceptual units. During this process, we removed any duplicated or similar items from the data. Ultimately, the data was organized into 14 functional units and 32 units for improvement. We finalized nine categories through the categorization of these units. Among them, seven items accounted for more than 10% of the additional requirements collected. Based on these results, we conducted feedback on each unit to identify any exceptional cases in the methodology.

This was a strategic goal that was set from the outset of this study's methodology review. When it comes to requirements engineering, the key priorities are feasibility, mutual consultation, and refinement in the end. The technique employed in this study

systematically considered items that are usually excluded from the feasibility study process due to a lack of understanding of the domain.

In Table 1, there are six modules with an additional collection rate of more than 10%. Among them, the Lab Grouping module has a high rate of 21.9%, and the Lab Management module has 19%.

**Table 1.** The result of the content analysis.

| Category | Module | Functional Requirement | Number of Functional Requirements | Number of Requirements Collected after Modeling | Percentage of Additional Requirements Collected |
|---|---|---|---|---|---|
| Professor/ System Administrator | VFT Lecture Management | Course Information, Dashboard, Announcements, Lecture Questions | 41 | 4 | 8.888889 |
| | Lab Grouping | Team Assignment, Team Tasks | 32 | 9 | 21.95122 |
| | Lab Participation Management | Input Participation for Each Lab, View Participation History | 23 | 5 | 17.85714 |
| | Lab Management | Field Trip Course Registration, Course Management | 17 | 4 | 19.04762 |
| Professor/Student | Lab Report | Report Registration, Report Management | 36 | 5 | 12.19512 |
| | Activity Management | Management of Offline Classroom Activities | 23 | 5 | 17.85714 |
| | Feedback Management | Providing Feedback to Students | 19 | 4 | 17.3913 |
| | VFT Lab Subject Management | Lab Subject, Lab Location, Lab Material Management | 26 | 6 | 18.75 |
| | VR Lab Management | Management of Labs Using VR, Device Management, Location Management | 37 | 4 | 9.756098 |

Examining the collected requirements, there are specifications that are difficult to identify through general requirement analysis. For example, the "short explanation" feature that occurs during a VFT session. While all students participate in the VFT, there are differences in knowledge among them, and sometimes the terminology can be challenging. In an offline class, students could ask questions to their neighbors to resolve this. However, this is more challenging in online labs. Therefore, requirements have been identified for features that systemically support these issues. Such requirements are difficult to discover in typical system requirement analyses. General requirement analysis often collects information from users before the system is implemented. Moreover, once the system has been developed, it is difficult to make improvements involving extensive structural changes. Therefore, it is easier to identify such requirements within a platform where multiple VFT modules are used simultaneously. This finding closely aligns with the objectives of this study.

For instance, the analysis of the attendance exception handling aspect of VFT in this experiment was not originally identified in the existing needs analysis, and the analysis of the group management aspect of VFT was only acknowledged in terms of quantitative student grouping in the existing needs analysis. This methodology introduced additional features, including grouping and group leader management, customized to the unique requirements of individual students based on their learning styles, interests, and preferred

subjects. From a methodological standpoint, statistical analysis has shown that over 20% of additional feature analyses were feasible in the student grouping function unit of the VFT module. A large portion of these analyzed features were deemed appropriate for the online field training platform.

A distinctive feature of the VFT platform proposed in this study is its ability to continuously improve through systematic feedback. This is highly useful in tasks where experience is valued and in unformalized work processes. Even when conducting field trips in offline classes, there are many teaching styles depending on the professor. Professors can offer higher-quality lectures to students as they accumulate field trip teaching experience. Similarly, VFTs have more limitations than offline field trips. As professors gain experience, each one develops their own expertise. The core of this study is to systematically manage the accumulated expertise of professors and explore optimal VFT lecture methods. The VFT platform contributes to substantially improving both the students' and educators' educational experience through this process.

To implement this technological ecosystem, it is necessary to establish a legacy for server-to-server interconnections. When connecting nodes of each server, it becomes challenging to consider the characteristics of the frameworks, and many server resources are consumed. Therefore, it is necessary to configure a central processing server and consider a vertical hierarchy structure for connections. The technological ecosystem is contemplated to be structured by utilizing a unified API service at the framework level on each server. VFTs implemented through this evolutionary modeling integrate each system holistically. Ultimately, a technological ecosystem, including a central processing server between VFTs, is formed. In this ecosystem, the central processing server processes and analyzes data in real-time to identify improvements. Therefore, introducing the MapReduce framework for data processing has system performance benefits in terms of intelligent methodologies in big data processing and component deployment. This framework can support parallel data analysis in a large-scale distributed computing environment like a VFT, which varies in characteristics depending on the subject and educational institution. We expect that technical research on open source frameworks that can properly integrate such individual characteristics will lead to higher student achievement.

The level of achievement in VFT depends on the user's active participation, due to the limitations of the platform. Problems arise when user participation is low due to factors like disconnection, functional errors, or switching lecture windows. While some of these issues can be partially addressed by integrating with a lecture video management system, for courses that require checking the overall operation at each experimental stage, like machinery or chemical experiments, visual support devices like VR must be introduced. It serves as a platform feature that substitutes for subjects where on-site learning is difficult, enabling students to have practical experiences.

In the current engineering education that involves advanced technologies, there is a large learning gap between universities due to limitations like the cost of experimental equipment. Additionally, there is often no online platform to substitute for students who used to receive offline engineering and vocational training, making education challenging. Further research is needed to explore platform technologies that can enhance educational achievements for students from the perspective of engineering and technology education.

This study aimed to address the functional complexity of the VFT platform from an evolutionary perspective and sought a methodology for designing a continually evolving structure. Within the platform, systems are interconnected, accumulating real-time data that functions as feedback within the platform. In the VFT curriculum, key actions and systematic support accumulate continuously in the system in the form of log data. The system provides a process through which it can technically advance while ensuring its unique activities. In this process, each module is continuously improved, and the entire platform learns from this feedback. As a result, a structure has been established where the platform evolves within the technological ecosystem. Empirical analyses were conducted

on the requirement analysis data analyzed through this methodology, and we confirmed that additional factors related to user behavior were identified.

The aim of this study was to suggest a more effective system design by incorporating technological and ecological elements, which refer to interactive functional components that various systems share within a technological ecosystem, into the educational information system. To facilitate the introduction of such intelligent systems, it is imperative to consider the educational and regulatory aspects as well. Moreover, there is a need for further reviews on supportive components for the proposed technological ecosystem in this study. This is because processing the data and analysis results generated from each node using a single method may hinder their advancement. In future studies, we aim to explore the introduction of a more intelligent methodology in big data processing and component deployment. The VFT formed through evolutionary modeling not only improves educational outcomes but also has advantages in collecting educational statistics. This is expected to greatly assist in the formulation of systematic educational policies, as well as support for educational administration and research.

First, in terms of educational outcomes, this platform allows for online practical education based on more objective and empirical data through the provision of student action data, learning statistics, and system improvement materials.

Second, from an educational administrative perspective, it supports multi-dimensional utilization and decision making of educational statistics. Educational administrative agencies and universities can revise their existing data collection methods through data requests and automated systems, thereby enhancing administrative efficiency.

**Author Contributions:** Conceptualization, H.H. and C.M.; methodology, C.M.; validation, C.M.; formal analysis, C.M.; investigation, H.H.; writing—original draft preparation, H.H. and C.M.; writing—review and editing, H.H. and C.M.; visualization, C.M.; supervision, C.M.; All authors have read and agreed to the published version of the manuscript.

**Funding:** This research was supported by Basic Science Research Program through the National Research Foundation of Korea (NRF) funded by the Ministry of Education (2020R1G1A1006677).

**Institutional Review Board Statement:** Not applicable.

**Informed Consent Statement:** Not applicable.

**Data Availability Statement:** Not applicable.

**Conflicts of Interest:** The authors declare no conflict of interest.

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
