# Peer review of "Evolutionary System Design for Virtual Field Trip Platform"

_sustainability, doi:10.3390/su152014917_

Round 1
Reviewer 1 Report
This paper suggest an effective system design by incorporating technological and ecological elements into the educational information system. In order to implement such intelligent virtual systems, it is imperative to consider the educational and regulatory aspects as well. Moreover, the data processing and analysis results generated from each node using a single method may hinder their advancement.
The paper provides an initial idea without emphasizing on the need of such virtual field trip framework. The standardization may be a problem but it is not useful until the learning outcomes are not known. It is true that during COVID-19 pandemic, such educational trips were postponed and students were not able to visit factories and industries to practically see the processes and manufacturing. However, it is known that many such field trips lost significance if they are not physically attended.
The first point to consider is that virtual field trips abbreviation should be VTF or VFT? Sometimes, VFT is used while at other places VTF is mentioned in the paper.
The need to adopt to intelligent systems in education is already investigated by many researchers. The use of virtual instructions and virtual labs is already there. Virtual field trips were first introduced in 2002. Since then, a number of research ideas were under consideration. Some of these are:
Makransky, G., & Mayer, R. E. (2022). Benefits of taking a virtual field trip in immersive virtual reality: Evidence for the immersion principle in multimedia learning. Educational Psychology Review, 34(3), 1771-1798.
Cheng, K. H., & Tsai, C. C. (2019). A case study of immersive virtual field trips in an elementary classroom: Students’ learning experience and teacher-student interaction behaviors. Computers & Education, 140, 103600.
Petersen, G. B., Klingenberg, S., Mayer, R. E., & Makransky, G. (2020). The virtual field trip: Investigating how to optimize immersive virtual learning in climate change education. British Journal of Educational Technology, 51(6), 2099-2115.
Mead, C., Buxner, S., Bruce, G., Taylor, W., Semken, S., & Anbar, A. D. (2019). Immersive, interactive virtual field trips promote science learning. Journal of Geoscience Education, 67(2), 131-142.
Keeping in view the existing work, it is not enough to propose a VFT framework without presenting (at least) a functional VFT system use case. In the literature, ideas have gone far beyond the virtual field trips and now working systems based on industrial metaverse will be soon integrated in the curriculum of scientific, engineering and techology courses. Despite that, physical field trips can not be replaced without a concrete reason by VFTs.
Moderate corrections are needed.
Author Response
Dear Reviewer Thank you for the opportunity to revise our manuscript, Evolutionary System Design for Virtual Field Trip Platform. We appreciate the careful review and constructive suggestions. It is our belief that the manuscript is substantially improved after making the suggested edits. Based on your guidance, we have tried our best to completely address your comments. Our replies (in RED) are below. The revisions are colored in RED in our revised manuscript. We hope this version of our submission of the manuscript is now acceptable and fulfils the expectations of the reviewer. Comments : Reviewer's Comment #1 : This paper suggest an effective system design by incorporating technological and ecological elements into the educational information system. In order to implement such intelligent virtual systems, it is imperative to consider the educational and regulatory aspects as well. Moreover, the data processing and analysis results generated from each node using a single method may hinder their advancement. -> Author's answer : We fully agree with your opinion. To implement an intelligent virtual system, it is essential to consider it from the perspective of an ecosystem composed of diverse elements, which has been reflected in the manuscript. > Revision in Manuscript: Ultimately, just as information systems are defined as systems that interact to provide the information needed to support organizations [15], educational information systems should be able to offer integrated education from the viewpoint of a virtual reality ecosystem aimed at sustainability. The virtual reality ecosystem consists of content, platform, network, and devices [16]. Reviewer's Comment #2: The paper provides an initial idea without emphasizing on the need of such virtual field trip framework. The standardization may be a problem but it is not useful until the learning outcomes are not known. It is true that during COVID-19 pandemic, such educational trips were postponed and students were not able to visit factories and industries to practically see the processes and manufacturing. However, it is known that many such field trips lost significance if they are not physically attended. -> Author's answer : We agree with the reviewer's opinion. In the case of VFT, visiting actual factories or industrial sites often yields higher learning effects. In this study, we acknowledge this limitation and propose a method to assist classes as closely aligned to real field trips as possible. This study conducted empirical analyses in interactive hands-on courses, such as foundational subjects in computer science and software education. We added content to the manuscript to explore additional ways VFT can aid in engineering education, addressing both limitations and directions for further research. > Revision in Manuscript: The level of achievement in VFT depends on the user's active participation, due to the limitations of the platform. Problems arise when the user participation is low due to factors like disconnection, functional errors, or switching lecture windows. While some of these issues can be partially addressed by integrating with a lecture video management system, for courses that require checking the overall operation at each experimental stage, like machinery or chemical experiments, visual support devices like VR must be introduced. It serves as a platform feature that substitutes for subjects where on-site learning is difficult, enabling students to have practical experiences. In the current engineering education that involves advanced technologies, there is a large learning gap between universities due to limitations like the cost of experimental equipment. Additionally, there is often no online platform to substitute for students who used to receive offline engineering and vocational training, making education challenging. Further research is needed to explore platform technologies that can enhance educational achievements for students from the perspective of engineering and technology education. Reviewer's Comment #3 : The first point to consider is that virtual field trips abbreviation should be VTF or VFT? Sometimes, VFT is used while at other places VTF is mentioned in the paper. The need to adopt to intelligent systems in education is already investigated by many researchers. The use of virtual instructions and virtual labs is already there. Virtual field trips were first introduced in 2002. Since then, a number of research ideas were under consideration. Some of these are: -> Author's answer : We have identified the mistakes after reviewing your comments. Thank you for pointing out the errors. We have changed all instances and figures containing "VTF" to "VFT." Reviewer's Comment #4: Keeping in view the existing work, it is not enough to propose a VFT framework without presenting (at least) a functional VFT system use case. In the literature, ideas have gone far beyond the virtual field trips and now working systems based on industrial metaverse will be soon integrated in the curriculum of scientific, engineering and techology courses. Despite that, physical field trips can not be replaced without a concrete reason by VFTs. -> Author's answer : We fully agree with your opinion. We have revised the introduction based on the following papers. > Revision in Manuscript: Meanwhile, there have been ongoing studies attempting to apply virtual reality in educational settings. Makransky and Mayer (2022) confirmed that immersive technologies using head-mounted displays make education feel lifelike, drawing students into enjoyment and interest, and leading to a positive learning experience [11]. When immersive virtual field trips (VFT) are conducted based on appropriate educational design, they can induce intended behavioral changes in students [12]. Additionally, research targeting elementary school students showed that immersive VFTs can help reduce test anxiety [13]. Furthermore, in the study by Meed, et al. (2019), adaptive feedback was integrated into VFTs to automatically provide the necessary information for individualized learning [14].
Reviewer 2 Report
The author has diligently provided detailed supplementary materials, which is commendable. However, the purpose and utilization of this data remain insufficiently explained. Necessary revisions are required.
1.There is insufficient reference to relevant literature, and the examples lack representativeness.
1-1.During the COVID-19 pandemic, many countries and regions implemented online distance learning. These instances provide ample reference materials for the author's review. For example, within the Asian region alone, there are numerous pertinent case studies published in MDPI academic journals, encompassing Japan, South Korea, and Taiwan.
1-2.Some references are not cited in the main text, while others mentioned in the main text are absent from the reference list. This aspect requires improvement.
2.The explanation of the research methodology is inadequate.
2-1.The connection between the proposed indicators and research hypotheses lacks thorough testing and validation.
2-2.The primary objective of this paper should be explicitly clarified to offer educational opportunities for professional readers.
3.The conclusion is excessively concise and does not align with the anticipated outcomes posited in the research hypotheses. Section Four's conclusion and discussion suffer from a shortage of adequate and cohesive data.
4.The comprehension of the research problem in this paper appears disjointed from the design and analysis of the research framework, posing difficulties in establishing an effective connection. This article highlights substantial structural issues requiring comprehensive rectification. Kindly implement the necessary revisions.
Author Response
Reviewer #1 Dear Reviewer Thank you for the opportunity to revise our manuscript, Evolutionary System Design for Virtual Field Trip Platform. We appreciate the careful review and constructive suggestions. It is our belief that the manuscript is substantially improved after making the suggested edits. Based on your guidance, we have tried our best to completely address your comments. Our replies (in RED) are below. The revisions are colored in RED in our revised manuscript. We hope this version of our submission of the manuscript is now acceptable and fulfils the expectations of the reviewer. Comments : Reviewer's Comment #1 : The author has diligently provided detailed supplementary materials, which is commendable. However, the purpose and utilization of this data remain insufficiently explained. Necessary revisions are required. There is insufficient reference to relevant literature, and the examples lack representativeness. Reviewer's Comment #1-1.During the COVID-19 pandemic, many countries and regions implemented online distance learning. These instances provide ample reference materials for the author's review. For example, within the Asian region alone, there are numerous pertinent case studies published in MDPI academic journals, encompassing Japan, South Korea, and Taiwan. -> Author's answer : We fully agree with your opinion. We have revised the content in the introduction accordingly. > Revision in Manuscript: Most educational activities in various Asian countries, including Hong Kong, Indonesia, and South Korea, have transitioned to online formats [6-8]. Offline field learning has either been canceled or switched to an instructor-centric online method [9]. Reviewer's Comment #1-2.Some references are not cited in the main text, while others mentioned in the main text are absent from the reference list. This aspect requires improvement. -> Author's answer : Thank you for your opinion. We have rechecked and revised the entire manuscript. We have added more references throughout the text. For example, the following sentences are included. > Revision in Manuscript : Various technologies drive the digital transformation of higher education institutions, depending on social, organizational, and technological perspectives [19]. LMS is categorized into open-source and commercial; in the case of open-source, it can be modified according to the educational institution's needs to reflect the organization's unique characteristics [22]. Le (2022) comparatively analyzed recorded and real-time methods like Zoom in terms of academic performance. The findings suggest that students in the lower 50% performed relatively worse in recorded classes [25]. Reviewer's Comment #2-1. The explanation of the research methodology is inadequate. The connection between the proposed indicators and research hypotheses lacks thorough testing and validation. -> Author's answer : We agree with the reviewer's opinion. The manuscript includes content (functions) such as ecosystems, requirements engineering, application of RFT system policies, and application of experiential activities as small-scale subjects. However, there is a lack of content regarding the achievement of research objectives through these functions. We have supplemented this part by adding content to the Research Method and Conclusion sections. > Revision in Manuscript: The level of achievement in VFT depends on the user's active participation, due to the limitations of the platform. Problems arise when the user participation is low due to factors like disconnection, functional errors, or switching lecture windows. While some of these issues can be partially addressed by integrating with a lecture video management system, for courses that require checking the overall operation at each experimental stage, like machinery or chemical experiments, visual support devices like VR must be introduced. It serves as a platform feature that substitutes for subjects where on-site learning is difficult, enabling students to have practical experiences. In the current engineering education that involves advanced technologies, there is a large learning gap between universities due to limitations like the cost of experimental equipment. Additionally, there is often no online platform to substitute for students who used to receive offline engineering and vocational training, making education challenging. Further research is needed to explore platform technologies that can enhance educational achievements for students from the perspective of engineering and technology education. Software architecture is not a fixed entity; it includes methods that are continuously refined to achieve the desired outcome. In software engineering, the advantages of an evolutionary process lie in its capacity for iterative improvements and speed. The greatest risk factor in software development is the need for large-scale modifications during the final stages. Evolutionary modeling has the attributes of 'immediate reflection of iterative improvements' and 'capability to utilize prototypes for decision-making.' Over time, this approach enables the development of software that meets user satisfaction by continually improving its features. One drawback of this modeling approach is that it involves iterative development phases, making it challenging to estimate the overall resources that will be invested in the development. However, its utility is high in continuously improving the quality of the end product. This modeling approach proposes a comprehensive technological ecosystem that integrates various VFT systems and EMIS, allowing these systems to function autonomously and accumulate performance data. Each system evolves its features based on this accumulated data. Evolutionary modeling is incorporated into software architecture. This architecture adapts to continually changing environments while maintaining functional stability. Microservice architecture has also been introduced from this perspective and has evolved into a methodology with high scalability when combined with the structural characteristics of system engineering in the technological ecosystem. Additionally, in the case of large international airport systems, there is a process that reflects historical data and execution outcomes in the system management and medium- to long-term planning. Through this process, a virtuous cycle is systematically established, leading to ongoing improvements. Reviewer's Comment #2-2. The primary objective of this paper should be explicitly clarified to offer educational opportunities for professional readers. -> Author's answer : We agree with the reviewer's opinion. The purpose of this study is to design a module that identifies the direction in which VFT functions are gradually evolving on a single platform. To technically review this, the procedure for integrating VFT functions into a single platform has been included. From this evolutionary perspective, the platform undergoes a process in which it ensures that each system can technically advance while maintaining its unique activities. In this process, each school's VFT processes continuously improve, and the platform learns this data to ensure the technological ecosystem functions progressively. From this viewpoint, we have supplemented the manuscript in the Method section. > Revision in Manuscript: The manuscript added here is the same as the manuscript added in Reviewer's Comment #2-2. We agree with the reviewer's opinion. The manuscript includes content (functions) such as ecosystems, requirements engineering, application of RFT system policies, and application of experiential activities as small-scale subjects. However, there is a lack of content regarding the achievement of research objectives through these functions. We have supplemented this part by adding content to the Research Method and Conclusion sections. > Revision in Manuscript: The level of achievement in VFT depends on the user's active participation, due to the limitations of the platform. Problems arise when the user participation is low due to factors like disconnection, functional errors, or switching lecture windows. While some of these issues can be partially addressed by integrating with a lecture video management system, for courses that require checking the overall operation at each experimental stage, like machinery or chemical experiments, visual support devices like VR must be introduced. It serves as a platform feature that substitutes for subjects where on-site learning is difficult, enabling students to have practical experiences. In the current engineering education that involves advanced technologies, there is a large learning gap between universities due to limitations like the cost of experimental equipment. Additionally, there is often no online platform to substitute for students who used to receive offline engineering and vocational training, making education challenging. Further research is needed to explore platform technologies that can enhance educational achievements for students from the perspective of engineering and technology education. Software architecture is not a fixed entity; it includes methods that are continuously refined to achieve the desired outcome. In software engineering, the advantages of an evolutionary process lie in its capacity for iterative improvements and speed. The greatest risk factor in software development is the need for large-scale modifications during the final stages. Evolutionary modeling has the attributes of 'immediate reflection of iterative improvements' and 'capability to utilize prototypes for decision-making.' Over time, this approach enables the development of software that meets user satisfaction by continually improving its features. One drawback of this modeling approach is that it involves iterative development phases, making it challenging to estimate the overall resources that will be invested in the development. However, its utility is high in continuously improving the quality of the end product. This modeling approach proposes a comprehensive technological ecosystem that integrates various VFT systems and EMIS, allowing these systems to function autonomously and accumulate performance data. Each system evolves its features based on this accumulated data. Evolutionary modeling is incorporated into software architecture. This architecture adapts to continually changing environments while maintaining functional stability. Microservice architecture has also been introduced from this perspective and has evolved into a methodology with high scalability when combined with the structural characteristics of system engineering in the technological ecosystem. Additionally, in the case of large international airport systems, there is a process that reflects historical data and execution outcomes in the system management and medium- to long-term planning. Through this process, a virtuous cycle is systematically established, leading to ongoing improvements. Reviewer's Comment #3.The conclusion is excessively concise and does not align with the anticipated outcomes posited in the research hypotheses. Section Four's conclusion and discussion suffer from a shortage of adequate and cohesive data. -> Author's answer : We agree with the reviewer's opinion. To enhance Section 4, we have restructured the paper as a whole. We have also supplemented empirical data to support the research findings. As the reviewer pointed out, there were content deficiencies in the Conclusion section. By adding descriptions about the research objectives, empirical analysis, user actions, future technical considerations, and contributions of the system, the structure of the methodology became much more accurate. We are very grateful for the excellent advice. Following this feedback, we proofread the Method section to improve the manuscript. > Revision in Manuscript: This study aimed to address the functional complexity of the VFT platform from an evolutionary perspective and sought a methodology for designing a continually evolving structure. Within the platform, systems are interconnected, accumulating real-time data that functions as feedback within the platform. In the VFT curriculum, key actions and systematic support accumulate continuously in the system in the form of log data. The system provides a process through which it can technically advance while ensuring its unique activities. In this process, each module is continuously improved, and the entire platform learns from this feedback. As a result, a structure has been established where the platform evolves within the technological ecosystem. Empirical analyses were conducted on the requirement analysis data analyzed through this methodology, and we confirmed that additional factors related to user behavior were identified. To perform a technical analysis of evolutionary modeling, we introduced content analysis to the existing system requirement analysis results. This method is a qualitative research approach from the perspective of instructors; its greatest advantage is that it allows for statistical analysis of the text data collected. The process was conducted in two stages. In the first stage, we analyzed the requirements collected for general EMIS and LMS. The second stage involved gathering requirements from feedback generated from an evolutionary modeling perspective after its application. We added explanations about the nature of the questions and the content analysis. Through this description, the justification for the methodology has been further strengthened. The four criteria for collecting requirement analysis data are as follows. Criterion 1: Is this feature similar to the educational effectiveness of a field trip? Y/N (Y: Q.2a, N: Q.2b) Criterion 2.a: What is the most similar feature between FT and VFT in terms of effectiveness? Criterion 2.b: What is the feature that most needs improvement in VFT? Criterion 3: What additional features are needed for application across various learning processes? Criterion 4: What features have been identified through the evolutionary functionality of VFT? We refined the analysis procedures in accordance with the definitions of content analysis for the requirement data. Based on the techniques of content analysis for requirement analysis, we refined the information corresponding to qualitative data. Next, during the sense-making process, we divided the data into units and categorized them according to core consistencies and meanings, generating descriptive evaluation indicators. Through this process, we are able to analyze the inherent characteristics of this methodology.We divided the requirement analysis text into sentences, performed pre-processing, and carried out a structured content analysis on the data. We analyzed all the clauses in the analytical data and identified 286 characteristics, 254 basic functions, and 63 areas for improvement. After conducting a pre-test on the result data, we removed any outliers. We categorized the data by core modules and conceptual units. During this process, we removed any duplicated or similar items from the data.Ultimately, the data was organized into 14 functional units and 32 units for improvement. We finalized nine categories through the categorization of these units. Among them, seven items accounted for more than 10% of the additional requirements collected. Based on these results, we conducted feedback on each unit to identify any exceptional cases in the methodology. In Table 1, there are six modules with an additional collection rate of more than 10%. Among them, the Lab Grouping module has a high rate of 21.9%, and the Lab Management module has 19%. Examining the collected requirements, there are specifications that are difficult to identify through general requirement analysis. For example, the "short explanation" feature that occurs during a VFT session. While all students participate in the VFT, there are differences in knowledge among them, and sometimes the terminology can be challenging. In an offline class, students could ask questions to their neighbors to resolve this. However, this is more challenging in online labs. Therefore, requirements have been identified for features that systemically support these issues. Such requirements are difficult to discover in typical system requirement analyses. General requirement analysis often collects information from users before the system is implemented. Moreover, once the system has been developed, it is difficult to make improvements involving extensive structural changes. Therefore, it is easier to identify such requirements within a platform where multiple VFT modules are used simultaneously. This finding closely aligns with the objectives of this study. A distinctive feature of the VFT platform proposed in this study is its ability to continuously improve through systematic feedback. This is highly useful in tasks where experience is valued and in unformalized work processes. Even when conducting field trips in offline classes, there are many teaching styles depending on the professor. Professors can offer higher-quality lectures to students as they accumulate field trip teaching experience. Similarly, VFTs have more limitations than offline field trips. As professors gain experience, each one develops their own expertise. The core of this study is to systematically manage the accumulated expertise of professors and explore optimal VFT lecture methods. The VFT platform contributes to substantially improving both the students' and educators' educational experience through this process. To implement this technological ecosystem, it is necessary to establish a legacy for server-to-server interconnections. When connecting nodes of each server, it becomes challenging to consider the characteristics of the frameworks, and many server resources are consumed. Therefore, it is necessary to configure a central processing server and consider a vertical hierarchy structure for connections. The technological ecosystem is contemplated to be structured by utilizing a unified API service at the framework level on each server. VFTs implemented through this evolutionary modeling integrate each system holistically. Ultimately, a technological ecosystem, including a central processing server between VFTs, is formed. In this ecosystem, the central processing server processes and analyzes data in real-time to identify improvements. Therefore, introducing the MapReduce framework for data processing has system performance benefits in terms of intelligent methodologies in big data processing and component deployment. This framework can support parallel data analysis in a large-scale distributed computing environment like a VFT, which varies in characteristics depending on the subject and educational institution. We expect that technical research on open-source frameworks that can properly integrate such individual characteristics will lead to higher student achievement. The VFT formed through evolutionary modeling not only improves educational outcomes but also has advantages in collecting educational statistics. This is expected to greatly assist in the formulation of systematic educational policies, as well as support for educational administration and research. First, in terms of educational outcomes, this platform allows for online practical education based on more objective and empirical data through the provision of student action data, learning statistics, and system improvement materials. Second, from an educational administrative perspective, it supports multi-dimensional utilization and decision-making of educational statistics. Educational administrative agencies and universities can revise their existing data collection methods through data requests and automated systems, thereby enhancing administrative efficiency. Reviewer's Comment #4.The comprehension of the research problem in this paper appears disjointed from the design and analysis of the research framework, posing difficulties in establishing an effective connection. This article highlights substantial structural issues requiring comprehensive rectification. Kindly implement the necessary revisions. -> Author's answer : We are thankful for and agree with the reviewer's opinion. We added the process of describing the modeling of the VFT research framework on a process-level basis to the Research Method section. We provided a detailed presentation of the functional flow for each process and added explanations about the educational objectives of the research model in the system analysis. These educational objectives were written to be applicable to the new educational environments where the introduction of VFT is required. Through this, the persuasiveness of the paper has been further strengthened. Additionally, we made two other considerations. First, we evaluated whether the research methodology of this study is suitable for the application of the evolving platform. Second, we considered the technical utility of system legacy and information processes within the platform. We have enhanced the manuscript from this perspective. As a result, we were able to more comprehensively substantiate the academic value of the research framework. We are very grateful to the reviewer. > Revision in Manuscript: Meanwhile, there have been ongoing studies attempting to apply virtual reality in educational settings. Makransky and Mayer (2022) confirmed that immersive technologies using head-mounted displays make education feel lifelike, drawing students into enjoyment and interest, and leading to a positive learning experience [12]. When immersive virtual field trips (VFT) are conducted based on appropriate educational design, they can induce intended behavioral changes in students [13]. Additionally, research targeting elementary school students showed that immersive VFTs can help reduce test anxiety [14]. Furthermore, in a study by Meed, et al. (2019), adaptive feedback was integrated into VFTs to automatically provide the necessary information for individualized learning [15]. Moreover, educational applications using augmented reality have yielded excellent results in terms of the quantity, complexity, and reliability of activities, compared to the control group, when tailored to students' levels of knowledge [16]. Examples of statistical experiential information are as follows. Strousopoulos, et al. (2023) developed a mobile application system for displaying sculptures and performed content recommendations using fuzzy weighting [27].

Reviewer 3 Report
Please find attached my comments

Moderate editing of English language required
Author Response
Sustainability revision round-1 - reviewer #3 Dear Reviewer Thank you for the opportunity to revise our manuscript, Evolutionary System Design for Virtual Field Trip Platform. We appreciate the careful review and constructive suggestions. It is our belief that the manuscript is substantially improved after making the suggested edits. Based on your guidance, we have tried our best to completely address your comments. Our replies (in RED) are below. The revisions are colored in RED in our revised manuscript. We hope this version of our submission of the manuscript is now acceptable and fulfils the expectations of the reviewer. Comments : The purpose of this study is to propose a systematic design methodology for supporting field trips in software education. The study signed a platform that integrates the VTF functions of each school's educational system into one, which guarantees each system's unique activities and supports the course of technological progression. This proposes a structure in which these processes continue to evolve, and the platform learns from this data to facilitate the evolution of the technological ecosystem. The paper discusses a hot topic of the related literature that the reader of this Journal would like to read. While this is a very interesting paper, I think it is necessary to address some concerns before publication. Some improvements should be done for a better comprehensive reading. Also, the following issues should be improved: Reviewer's Comment #1. In the introduction section while you reference various technological advancements and their impacts, it would be beneficial to provide specific examples or data to support your claims. For instance, when discussing the challenges faced during the COVID-19 pandemic (paragraph starting with "Despite the development of these technologies..."), you can include statistics or references to studies highlighting the difficulties faced by students and the limitations of the Learning Management System (LMS). Providing concrete evidence strengthens your argument. -> Author's answer : We appreciate the reviewer’s comment. We fully agree with your opinion. We recognized that the previous sentences leading into the introduction were not logical. Therefore, we focused on the platform within the virtual environment ecosystem to align with the main content of our paper. > Revision in Manuscript: The virtual reality ecosystem consists of content, platforms, networks, and devices[17]. Previous studies[12-16] have focused on educational content and devices. In contrast, our study focuses on a platform that connects various stakeholders online and allows for the supply of educational content by integrating existing legacy systems. Reviewer's Comment #2. Ensure consistency in terminology and concepts. For example, you mention "evolutionary mechanisms" (line 30) and "evolutionary education system" (line56). It would be helpful to clarify or define these terms for readers. -> Author's answer : We agree. To address the reviewer's comments regarding the lack of terminology definitions, we have proofread the manuscript. We have supplemented descriptions for terms like "evolutionary mechanisms," "evolutionary education system," "evolutionary modeling," "evolutionary methodology," "evolutionary method," "evolutionary element," "ecological elements," and "technological ecosystem." Through this, we improved the overall quality of the manuscript. > Revision in Manuscript: 1) "evolutionary mechanisms": Evolutionary mechanisms refer to a methodology aimed at maintaining progressive design within a system. It involves a process of iterative improvements according to formalized analysis procedures, using diagrammatic tools. 2) "evolutionary education system": An evolutionary education system is a system that has adopted evolutionary modeling techniques in the educational system. It takes the form of a prototype that continuously reflects improvements in the ongoing communication processes with students. 3) "evolutionary modeling": Evolutionary modeling is a representative form of system design methodology. 4) "evolutionary methodology": In this manuscript, the term evolutionary methodology is used in a manner similar to evolutionary modeling. Therefore, it was combined with evolutionary modeling. 5) "evolutionary method": The evolutionary method integrates and organizes the attributes of evolutionary elements in terms of their interrelatedness. 6) "evolutionary element": An evolutionary element refers to an individual feature that engages in progressive interactions within the system. The evolutionary method is the functionalization of this element within an independent object. 7) "ecological elements": Ecological elements refer to interactive functional components that various systems share within a technological ecosystem. 8) "technological ecosystem": A technological ecosystem refers to the structure formed by all systems installed within a platform, each performing its own independent functions. Within this technological ecosystem, each system maintains interdependence and functional completeness. Reviewer's Comment #3. Consider providing a brief overview or roadmap of the main points you will discuss in the subsequent sections of your paper. This can help readers anticipate the content and structure of your work. -> Author's answer : We are very grateful for the excellent advice. We had focused on describing prior research directly related to this study and the research objectives in the introduction. As a result, the explanation of logical flow from the perspective of system methodology was lacking. This study deals extensively with processes and flows, much like evolutionary processes. Therefore, it is deemed essential to provide an overview and roadmap in the introduction of the paper. We have more thoroughly explained the procedures of the methodology and strengthened the logic of the methodology. > Revision in Manuscript: In this study, we propose a VFT design methodology following the flow outlined below, aiming to explore the evolutionary perspective in the field of information system software. 1) In the Introduction section, we first discuss system development methodology. We present the development process and functional changes of online education systems. We explore approaches for delivering learning experiences in the system to achieve educational goals. 2) In the Method section, we propose more precise methodology design techniques to systematically support experiential learning for students. The platform operates as a single technological ecosystem, and the system undergoes continuous development to sustain educational activities. We analyze the interactions of data emerging from these processes and incorporate the findings into the system. Within the framework of platform policies, this cycle repeats, and the system evolves through interactions with other systems. 3) In the Results section, we compare the VFT methodology with existing software design methodologies. Through usability tests conducted with five users, we analyzed the evolutionary elements and characteristics each system possesses within the technological ecosystem. From this perspective, we analyze the characteristics of the VFT methodology. 4) In the Conclusion section, we conducted an empirical analysis. As a result, we confirmed that user behavioral elements were additionally derived by more than 20% in the features designed using the VFT methodology. Reviewer's Comment #4. While you've mentioned some prior research and sources, consider engaging with them more deeply by discussing how your proposed research or findings relate to or build upon existing literature. -> Author's answer : Thank you for your opinion. We found that previous research focused more on education-centric content and devices. From an ecosystem perspective, we believe that the role of the platform is equally important as content and devices. > Revision in Manuscript: The platform evolves, and its features continually reconfigure through systematic feedback. For example, even in a basic web board with just four fundamental functions—input, edit, delete, and update—refined features are updated through various user characteristics. These updates accumulate and lead to the development of a unique system for the platform. Additionally, it is not just one element in the system that evolves independently; the development of each element is considered alongside system requirements in terms of content, network, and devices to have a positive impact on the overall platform. Therefore, from the perspective of overall components within the platform, we can keep pace with the speed of technological advancements. Reviewer's Comment #5. In section 2, the authors mention statistics and figures (e.g., the LMS market size and growth rate), but it lacks specific citations or references to support these claims. It's essential to provide the sources of such data to ensure academic rigor. Additionally, when referencing phases of education during COVID-19 (e.g.,"disruption," "transition," and "reimagining"), include citations to relevant studies or reports that discuss these phases in detail. -> Author's answer : We agree with your opinion. We have taken content from the report and properly reflected the accurate sources. Additionally, we cited appropriate examples concerning interruption, transition, and re-imagination. > Revision in Manuscript: The outbreak of COVID-19 caught everyone off guard. However, much like how software development outcomes differ depending on the level of capability maturity model integration (CMMI) [21], organizations showed significant differences in their responses based on the maturity level of their digital technology. For instance, despite being willing to participate in online classes, students still showed a preference for face-to-face classes or felt that they were inferior [22]. Since participation is a key determinant in learning, we made various attempts to offer students virtual real-world experiences, museum and gallery tours, simulations, and sandbox environments [20]. Reviewer's Comment #6. Provide a clear context for your study by explaining why the topics discussed in the introduction are relevant and what knowledge gaps or research questions your study aims to address. -> Author's answer : We fully agree with your opinion. In the introduction, we noticed that the content explaining why this research is necessary was insufficient. We have corrected this accordingly. > Revision in Manuscript: Meanwhile, there have been ongoing studies attempting to apply virtual reality in educational settings. Makransky and Mayer (2022) confirmed that immersive technologies using head-mounted displays make education feel lifelike, drawing students into enjoyment and interest, and leading to a positive learning experience [12]. When immersive virtual field trips (VFT) are conducted based on appropriate educational design, they can induce intended behavioral changes in students [13]. Additionally, research targeting elementary school students showed that immersive VFTs can help reduce test anxiety [14]. Furthermore, in a study by Meed, et al. (2019), adaptive feedback was integrated into VFTs to automatically provide the necessary information for individualized learning [15]. Reviewer's Comment #7. In research methodology section I read about references previous studies and statistics, but it lacks specific citations or references to support these claims. To ensure academic rigor, include citations to relevant sources or studies that provide evidence for the points made. For example, when discussing the evolution of the South Korean education system, provide references to relevant reports or sources. Since the literature review should cover all fields, and to support several assertions, the authors are advised to use the following references: -> Author's answer : We agree with your opinion. We have thoroughly reviewed the papers you recommended and incorporated them, as we believe they will enhance the academic theory and credibility of our paper. > Revision in Manuscript: Moreover, educational applications using augmented reality have yielded excellent results in terms of the quantity, complexity, and reliability of activities, compared to the control group, when tailored to students' levels of knowledge [16]. Examples of statistical experiential information are as follows. Strousopoulos, et al. (2023) developed a mobile application system for displaying sculptures and performed content recommendations using fuzzy weighting [27]. Reviewer's Comment #8. While the section provides technical details, it should aim for a balance between technical depth and accessibility. Some technical terms and acronyms are used without clear explanations, which may pose challenges for readers who are not well-versed in the subject matter. Ensure that specialized terms are defined or explained for a broader audience. -> Author's answer : We agree with the reviewer's opinion. Some technical terms and acronyms were used without definition. We have added explanations for terms like factory method pattern, DFD, iterator pattern, state transition, and EMIS to make them more accessible to readers. > Revision in Manuscript: 1) Factory method pattern: The factory method pattern is a design pattern in object-oriented programming that deals with the creation of specific classes. It allows the associated classes to determine which objects to create. 2) DFD: A data-flow diagram (DFD) is a diagram that represents how data transforms and flows through various processes within software. 3) Iterator pattern: The iterator pattern is a design pattern in software that separates the access functionality from the data structure within an object. 4) state transition: State transition represents the functional flow within a system based on states. 5) EMIS: Educational Management Information Systems (EMIS) are information systems used in schools, typically including the features of an academic management system. Reviewer's Comment #9. Provide a clear and concise introduction to the research methodology section that explains its significance in the broader context of the study. Explain why the chosen methodology is suitable for addressing the research objectives. -> Author's answer : We chose evolutionary modeling techniques to construct a technological ecosystem for systems. This evolutionary modeling approach proposes a comprehensive technological ecosystem that integrates various VFT systems and EMIS, allowing these systems to function autonomously and accumulate performance data. Each system evolves its features based on this accumulated data. Additional details for the manuscript are included in Item 10 below. Reviewer's Comment #10. Explain in more detail why an evolutionary modeling technique is chosen as the methodology and how it addresses the limitations of existing EISs. Provide a clear rationale for this choice. -> Author's answer : We chose evolutionary process modeling for this study because it allows for the 'immediate reflection of iterative improvements' and 'capability to utilize prototypes for decision-making.' Over time, this approach enables the development of software that meets user satisfaction by continually improving its features. VFT systems exhibit varying functionalities depending on the type of subject matter. This is a limitation of EMIS. We chose evolutionary modeling techniques to create a system that can assist students. > Revision in Manuscript: Software architecture is not a fixed entity; it includes methods that are continuously refined to achieve the desired outcome. In software engineering, the advantages of an evolutionary process lie in its capacity for iterative improvements and speed. The greatest risk factor in software development is the need for large-scale modifications during the final stages. Evolutionary modeling has the attributes of 'immediate reflection of iterative improvements' and 'capability to utilize prototypes for decision-making.' Over time, this approach enables the development of software that meets user satisfaction by continually improving its features. One drawback of this modeling approach is that it involves iterative development phases, making it challenging to estimate the overall resources that will be invested in the development. However, its utility is high in continuously improving the quality of the end product. This modeling approach proposes a comprehensive technological ecosystem that integrates various VFT systems and EMIS, allowing these systems to function autonomously and accumulate performance data. Each system evolves its features based on this accumulated data. Evolutionary modeling is incorporated into software architecture. This architecture adapts to continually changing environments while maintaining functional stability. Microservice architecture has also been introduced from this perspective and has evolved into a methodology with high scalability when combined with the structural characteristics of system engineering in the technological ecosystem. Additionally, in the case of large international airport systems, there is a process that reflects historical data and execution outcomes in the system management and medium- to long-term planning. Through this process, a virtuous cycle is systematically established, leading to ongoing improvements. References Reviewer's Comment #11. The "Conclusion and Discussion" section provides valuable insights into the research's empirical analysis and its impact on system design. However, there are areas that require major revisions to enhance clarity, coherence, and the depth of the discussion. Below, I outline the major revisions needed: a. The conclusion should begin with a clear restatement of the research objectives and a concise summary of the main findings. Currently, it is not immediately clear what the primary objectives were and what key insights were gained through empirical analysis. -> Author's answer : We agree with the reviewer's opinion. This paper discussed methodologies for platform design. However, the conclusion lacks detailed explanations on the effectiveness of these design methodologies. Therefore, we found out a lack of description on the value of this study. To address this, we have enhanced the Conclusion and Discussion section. > Revision in Manuscript: This study aimed to address the functional complexity of the VFT platform from an evolutionary perspective and sought a methodology for designing a continually evolving structure. Within the platform, systems are interconnected, accumulating real-time data that functions as feedback within the platform. In the VFT curriculum, key actions and systematic support accumulate continuously in the system in the form of log data. The system provides a process through which it can technically advance while ensuring its unique activities. In this process, each module is continuously improved, and the entire platform learns from this feedback. As a result, a structure has been established where the platform evolves within the technological ecosystem. Empirical analyses were conducted on the requirement analysis data analyzed through this methodology, and we confirmed that additional factors related to user behavior were identified. b. Expand on the discussion of the key findings and their implications for system design. Provide specific examples and results from the empirical analysis to illustrate how the design methodology addressed identified needs and challenges. -> Author's answer : We agree with the reviewer's opinion. This study focuses on the advantages that system analysts can discover through an evolutionary platform. Therefore, we analyzed the additional functional requirements collected through this platform. In this process, we were able to identify important requirements that were not readily apparent using conventional system analysis techniques. For example, consider the "short lecture" feature that occurs during a VFT session. While all students participate in the VFT, there are differences in knowledge among them, and sometimes the terminology can be challenging. In an offline lecture, students could easily ask questions to their neighbors to resolve such issues, but this is more difficult in an online setting. Therefore, a system feature that supports this aspect is necessary. Such requirements are hard to discover in standard system requirement analysis but are more easily found within platforms where multiple systems are interlocked and used together. This is the purpose of this research, highlighting the platform's innovative characteristics. We have added these requirements to the manuscript in the form of a table. > Revision in Manuscript: To perform a technical analysis of evolutionary modeling, we introduced content analysis to the existing system requirement analysis results. This method is a qualitative research approach from the perspective of instructors; its greatest advantage is that it allows for statistical analysis of the text data collected. The process was conducted in two stages. In the first stage, we analyzed the requirements collected for general EMIS and LMS. The second stage involved gathering requirements from feedback generated from an evolutionary modeling perspective after its application. We added explanations about the nature of the questions and the content analysis. Through this description, the justification for the methodology has been further strengthened. The four criteria for collecting requirement analysis data are as follows. Criterion 1: Is this feature similar to the educational effectiveness of a field trip? Y/N (Y: Q.2a, N: Q.2b) Criterion 2.a: What is the most similar feature between FT and VFT in terms of effectiveness? Criterion 2.b: What is the feature that most needs improvement in VFT? Criterion 3: What additional features are needed for application across various learning processes? Criterion 4: What features have been identified through the evolutionary functionality of VFT? We refined the analysis procedures in accordance with the definitions of content analysis for the requirement data. Based on the techniques of content analysis for requirement analysis, we refined the information corresponding to qualitative data. Next, during the sense-making process, we divided the data into units and categorized them according to core consistencies and meanings, generating descriptive evaluation indicators. Through this process, we are able to analyze the inherent characteristics of this methodology. We divided the requirement analysis text into sentences, performed pre-processing, and carried out a structured content analysis on the data. We analyzed all the clauses in the analytical data and identified 286 characteristics, 254 basic functions, and 63 areas for improvement. After conducting a pre-test on the result data, we removed any outliers. We categorized the data by core modules and conceptual units. During this process, we removed any duplicated or similar items from the data. Ultimately, the data was organized into 14 functional units and 32 units for improvement. We finalized nine categories through the categorization of these units. Among them, seven items accounted for more than 10% of the additional requirements collected. Based on these results, we conducted feedback on each unit to identify any exceptional cases in the methodology. c. The section mentions that empirical analysis led to the discovery of user behavioral factors and additional features. Elaborate on these factors and features, providing insights into how they were identified, why they are important, and how they were integrated into the design methodology. -> Author's answer : We agree with the reviewer's opinion. The current version of the conclusion lacks detailed empirical analysis results. Thus, the paper has the drawback of an incomplete conclusion. We have incorporated the reviewer's comments and added detailed empirical explanations to the conclusion. Additionally, we enhanced the content concerning the discussion aspect in the conclusion. Specifically, we included evaluations on areas for improvement within both the research methodology and analysis results. As a result, the logic of the paper has been strengthened. In Table 1, there are six modules with an additional collection rate of more than 10%. Among them, the Lab Grouping module has a high rate of 21.9%, and the Lab Management module has 19%. Examining the collected requirements, there are specifications that are difficult to identify through general requirement analysis. For example, the "short explanation" feature that occurs during a VFT session. While all students participate in the VFT, there are differences in knowledge among them, and sometimes the terminology can be challenging. In an offline class, students could ask questions to their neighbors to resolve this. However, this is more challenging in online labs. Therefore, requirements have been identified for features that systemically support these issues. Such requirements are difficult to discover in typical system requirement analyses. General requirement analysis often collects information from users before the system is implemented. Moreover, once the system has been developed, it is difficult to make improvements involving extensive structural changes. Therefore, it is easier to identify such requirements within a platform where multiple VFT modules are used simultaneously. This finding closely aligns with the objectives of this study. d. Discuss the relevance of the proposed system design and design methodology within the broader educational context. How might the identified factors and features contribute to improving the educational experience for students and educators? -> Author's answer : We agree with the reviewer's opinion. In the case of field trips or lab courses, the professor's teaching experience has a significant impact on the quality of the lecture. An evolutionary aspect of this system takes into account the accumulation of such experience. Moreover, we have considered the practical and academic contributions of this framework in relation to system design methodologies. The goal of this study is to smoothly apply evolutionary modeling methodologies to students' online practical education. This methodology has a strength in empirically analyzing student achievement. > Revision in Manuscript: A distinctive feature of the VFT platform proposed in this study is its ability to continuously improve through systematic feedback. This is highly useful in tasks where experience is valued and in unformalized work processes. Even when conducting field trips in offline classes, there are many teaching styles depending on the professor. Professors can offer higher-quality lectures to students as they accumulate field trip teaching experience. Similarly, VFTs have more limitations than offline field trips. As professors gain experience, each one develops their own expertise. The core of this study is to systematically manage the accumulated expertise of professors and explore optimal VFT lecture methods. The VFT platform contributes to substantially improving both the students' and educators' educational experience through this process. e. While the section mentions feasibility, it would be beneficial to discuss the practical feasibility of implementing the proposed design and features in real-world educational settings. Address any potential challenges or barriers to implementation. -> Author's answer : We agree with the reviewer's opinion. There are prerequisites to operate this platform across multiple school systems. An API server must be prepared between the systems of various schools, and server-to-server interconnection features must be implemented. Many school systems have their own unique frameworks, leading to substantial development costs to synchronize data between servers. As a solution, setting up a central processing server that provides a unified API service for each framework could be considered. > Revision in Manuscript: To implement this technological ecosystem, it is necessary to establish a legacy for server-to-server interconnections. When connecting nodes of each server, it becomes challenging to consider the characteristics of the frameworks, and many server resources are consumed. Therefore, it is necessary to configure a central processing server and consider a vertical hierarchy structure for connections. The technological ecosystem is contemplated to be structured by utilizing a unified API service at the framework level on each server. The level of achievement in VFT depends on the user's active participation, due to the limitations of the platform. Problems arise when the user participation is low due to factors like disconnection, functional errors, or switching lecture windows. While some of these issues can be partially addressed by integrating with a lecture video management system, for courses that require checking the overall operation at each experimental stage, like machinery or chemical experiments, visual support devices like VR must be introduced. It serves as a platform feature that substitutes for subjects where on-site learning is difficult, enabling students to have practical experiences. In the current engineering education that involves advanced technologies, there is a large learning gap between universities due to limitations like the cost of experimental equipment. Additionally, there is often no online platform to substitute for students who used to receive offline engineering and vocational training, making education challenging. Further research is needed to explore platform technologies that can enhance educational achievements for students from the perspective of engineering and technology education. f. Briefly mention the need for further reviews and exploration of intelligent methodologies in big data processing and component deployment. Expand on this by outlining specific areas for future research and the potential impact of such research on educational information systems. -> Author's answer : We agree with the reviewer's opinion. VFT includes the process of real-time data processing and analysis generated from various information systems to find feedback. Therefore, considerations are needed in terms of intelligent methodologies in big data processing and component deployment. Thus, the MapReduce framework for data processing must be introduced. This framework can support the parallel analysis of large volumes of data in a large-scale distributed computing environment like VFT. Additionally, VFT varies in characteristics depending on the subject and educational institution. In follow-up research, techniques for combining these individual characteristics into respective components are considered. > Revision in Manuscript: VFTs implemented through this evolutionary modeling integrate each system holistically. Ultimately, a technological ecosystem, including a central processing server between VFTs, is formed. In this ecosystem, the central processing server processes and analyzes data in real-time to identify improvements. Therefore, introducing the MapReduce framework for data processing has system performance benefits in terms of intelligent methodologies in big data processing and component deployment. This framework can support parallel data analysis in a large-scale distributed computing environment like a VFT, which varies in characteristics depending on the subject and educational institution. We expect that technical research on open-source frameworks that can properly integrate such individual characteristics will lead to higher student achievement. g. Summarize the key takeaways from the study and emphasize the contributions it makes to the field of educational information systems and technology. Clearly articulate the significance of the research in advancing system design practices. -> Author's answer : We agree with the reviewer's opinion. The conclusion only contained a brief description of the results, failing to demonstrate the importance and contributions of the research. To address this, the description was expanded, as suggested by the reviewer. We have more thoroughly explained the research value of this study and strengthened the logic of the methodology. Thank you very much. > Revision in Manuscript: The VFT formed through evolutionary modeling not only improves educational outcomes but also has advantages in collecting educational statistics. This is expected to greatly assist in the formulation of systematic educational policies, as well as support for educational administration and research. First, in terms of educational outcomes, this platform allows for online practical education based on more objective and empirical data through the provision of student action data, learning statistics, and system improvement materials. Second, from an educational administrative perspective, it supports multi-dimensional utilization and decision-making of educational statistics. Educational administrative agencies and universities can revise their existing data collection methods through data requests and automated systems, thereby enhancing administrative efficiency. Concluding, the structure of the paper is good, and the paper is well organized. However, major revisions should be considered prior to publication. -> Author's answer : We are sincerely grateful for your thorough consideration and scrutiny of our manuscript, “Evolutionary System Design for Virtual Field Trip Platform”. Through the accurate comments made by the reviewers, we better understand the critical issues in this paper. We have revised the manuscript according to the Reviewer’s suggestions. We hope that our revised manuscript will be considered and accepted for publication in the Sustainability.
Reviewer 4 Report
It is a very interesting idea whose applicability is very realistic and necessary. In the field of education, experimental activities and an active participation in education are necessary, to manage statistical information and create a history of these data.
It is a well structured study and the ideas are clearly expressed.
The purpose is highlighted and explained, which brings a note of depth in the organization of information and its applicability, very necessary in the field of education. Working models are necessary in education, and their choice by the students brings that necessary addition to knowledge.
The conclusions are clearly expressed.
This study can be a benchmark for those who want to apply the suggested methods.
Author Response
We are sincerely grateful for your thorough consideration and scrutiny of our manuscript, “Evolutionary System Design for Virtual Field Trip Platform”. Through the accurate comments made by the reviewers, we better understand the critical issues in this paper. We have revised the manuscript according to the Reviewer’s suggestions.
We hope that our revised manuscript will be considered and accepted for publication in the Sustainability.
Round 2
Reviewer 1 Report
The author's have improved the paper as per the suggested changes. VFT is getting popular for the online remote education. At a first place, it can be used to demonstrate students as a pre-visit briefing coupled with a physical on-site tour. Once immersive VFT is implemented, it can finally replace the physical presence and save time and expense as well offering remote visits to a number of key places.
Reviewer 2 Report
Accept the paper in its present form; the authors addressed the recommendations made.
Reviewer 3 Report
All my comments have been taken into consideration, congratulations on your effort!
Minor editing of English language required